# Mycosporine-Like Amino Acids: Making the Foundation for Organic Personalised Sunscreens

**DOI:** 10.3390/md17110638

**Published:** 2019-11-12

**Authors:** Nedeljka N. Rosic

**Affiliations:** 1School of Health and Human Sciences, Southern Cross University, Southern Cross Drive, Bilinga, QLD 4225, Australia; 2Marine Ecology Research Centre, Southern Cross University, Lismore, Military Rd, East Lismore, NSW 2480, Australia

**Keywords:** mycosporine-like amino acids, mycosporine-like amino acid biosynthesis, sunscreen, antioxidant, antiaging, anti-inflammatory, DNA protection, ultraviolet-absorbing compounds, cosmetics, mycosporine-like amino acid gene regulation

## Abstract

The surface of the Earth is exposed to harmful ultraviolet radiation (UVR: 280–400 nm). Prolonged skin exposure to UVR results in DNA damage through oxidative stress due to the production of reactive oxygen species (ROS). Mycosporine-like amino acids (MAAs) are UV-absorbing compounds, found in many marine and freshwater organisms that have been of interest in use for skin protection. MAAs are involved in photoprotection from damaging UVR thanks to their ability to absorb light in both the UV-A (315–400 nm) and UV-B (280–315 nm) range without producing free radicals. In addition, by scavenging ROS, MAAs play an antioxidant role and suppress singlet oxygen-induced damage. Currently, there are over 30 different MAAs found in nature and they are characterised by different antioxidative and UV-absorbing capacities. Depending on the environmental conditions and UV level, up- or downregulation of genes from the MAA biosynthetic pathway results in seasonal fluctuation of the MAA content in aquatic species. This review will provide a summary of the MAA antioxidative and UV-absorbing features, including the genes involved in the MAA biosynthesis. Specifically, regulatory mechanisms involved in MAAs pathways will be evaluated for controlled MAA synthesis, advancing the potential use of MAAs in human skin protection.

## 1. Introduction

Due to a reduction in aerosols and cloud cover, the levels of ultraviolet radiation (UVR) reaching the Earth’s surface are predicted to increase during the 21st century [1]. Organisms have developed several photoprotective mechanisms to survive high levels of UVR. Different mitigation strategies are utilised by different species and often in combination including DNA repair systems, antioxidant activities, and the application of UV-absorbing compounds. Exposure to UVR results in reactive oxygen species (ROS) production, oxidative stress and DNA damage. The mitigation strategies to reduce the UV-induced damage include DNA repair mechanisms via processes such as photoreactivation, excision, and mismatch repair. In response to the generation of ROS, organisms start accumulating antioxidants to capture free radicals. In addition, as a part of the UVR interception response, organisms accumulate photoprotective compounds with UVR absorption capabilities such as mycosporine-like amino acids (MAAs) [2,3,4,5]. MAAs are found in a large number of aquatic species, including marine and freshwater organisms that have been exposed to high levels of damaging UVR [6]. The vast variety of species containing MAAs includes phytoplankton, cyanobacteria, fungi, macroalgae, microalgae, as well as animals coming from both aquatic and terrestrial ecosystems [7]. These ubiquitous and highly abundant secondary metabolites are found to have a critical photoprotective role in these aquatic species [8]. MAAs have been evolutionarily conserved in aquatic organisms [9], together with some other important photoprotective compounds such as scytonemin in cyanobacteria [10,11], carotenoid pigments in plants and many microorganisms [12,13]. 

In this review, the aim was (1) to provide a summary of the MAA UV-absorbing, antioxidative, anti-inflammatory; antiaging features; (2) to discuss the genes involved in MAA biosynthesis and (3) to evaluate the regulatory mechanisms involved in MAA synthesis. In particular, this review assesses current knowledge about the biosynthesis of MAAs and suggests possible directions for the evolving biotechnological potential of MAAs in human skin protection.

## 2. MAA Diversity 

MAAs are a diverse group of colourless and hydrophilic compounds characterised by their small molecular mass (<400 Da). In their core, MAAs are composed of a cyclohexenone or a cyclohexenimine ring conjugated to an amino acid residue or its imino alcohol (Figure 1A) [14,15,16]. MAAs absorb mainly in the range of 310 to 362 nm and are characterised with high molar extinction coefficients (*ε* = 28,100 to 50,000 M^−1^ cm^−1^) [6,17]. There are over 30 different MAA compounds identified in nature [18,19] and probably more to be discovered with further development of novel high-throughput technologies. The examples of five major MAAs found in most aquatic species including their direct precursor, 4-deoxygadusol (4-DG), are presented in Figure 1B. 

A huge variety of MAAs have been reported in cyanobacteria [20,21,22], red algae [23,24,25,26,27,28], fungi [29], green algae [30,31], dinoflagellates [32,33,34], invertebrates (e.g., sponges [15,35,36], corals [21,37,38], sea urchins [39,40]) and vertebrates such as fish [41,42,43,44]. However, not all species have all the MAAs. From all organisms exposed to high levels of UV radiation, cyanobacteria have been under an especially huge evolutionary pressure to survive damaging UV radiation. An overview of MAAs reported in this diverse taxonomic group provides an important summary of the organisms’ ability to adapt for survival [22]. Red algae (Rhodophyta) are characterised by the highest diversity and concentrations of MAAs [16,45]. In a new study, MAA profiles were established for 23 red algae, with the most abundant MAAs being shinorine, palythine, asterina-330; porphyra-334 [28]. Furthermore, six new MAAs were recently chemically characterised in the red alga *Bostrychia scorpioides* [46]. In addition, two new MAAs (LC (*Lendenfeldia chondrode)*-343 and mycosporine-ethanolamine) along with well-known asterina-330 and shinorine were recently isolated from the marine sponge *Lendenfeldia chondrode* [36].

MAAs were also reported in many other animals (e.g., fish and sea stars) that naturally lack the MAA biosynthetic pathway, but do acquire MAAs through their algal diet or symbiosis with algae or/and bacterial symbionts [6,7,42,47]. However, beyond the diet, *de novo* synthesis of gadusol (the MAA precursor) was reported in some fish [43,44] and corals [48]. A wide range of MAAs was reported in the coral *Pocillopora capitate,* including both primary and secondary MAAs: mycosporine-glycine, shinorine, porphyra-334, mycosporine-methylamine-serine, mycosporine-methylamine-threonine, palythine-serine, palythine, and palythine-threonine [49]. In holobionts like corals, the presence of their microbial symbionts may be an efficient way to develop and adapt to different UV conditions [6,32]. However, coral dinoflagellates in cultures had less diverse MAA profiles compared to when dinoflagellates were in symbiosis within the coral host [32,49]. 

Identification and characterisation of MAAs requires the use of multiple diverse chemistry methods and measurements. The overview of different methodologies applied for the identification and characterisation of some common MAAs are provided in a very comprehensive review by Carreto and Carigan [16]. Furthermore, different methods of the MAA extractions are used for the successful isolation of MAAs. The appropriate methods for MAA extractions are influenced by the organisms’ characteristics and the tissue types used for extraction [45]. For the characterisation of MAAs, the combination of different methods is required, usually including the use of high-performance liquid chromatography (HPLC) and mass spectrometry (Figure 2, [45]). Multiple methods including HPLC analysis, reverse-phase liquid chromatography-mass spectrometry (RPLC-MS) and hydrophilic interaction liquid chromatography (HILIC) were recently used for the discovery and characterisation of novel MAAs in cyanobacteria [50]. The importance of combining different methods of chemical analyses characterised by improved sensitivity was recently confirmed by Lagegerie et al. [27]. The initial analyses of MAAs in red macroalgae from the Brittany region in France resulted in the detection of 23 potential MAAs using HPLC; further analyses using liquid chromatography–mass spectrometry (LC-MS) confirmed that only six different types of MAAs (shinorine, palythine, asterina-330, porphyra-334, usurijene; palythene) were found in 40 species of red macroalgae [27]. Additional and more comprehensive characterisation of the MAA chemical structures can be done by a combination of nuclear magnetic resonance (NMR) and LC-MS analyses [51]. In addition, the combination of infrared (IR) spectroscopic analysis and gas chromatography (GC)-MS analysis is also used for improving the characterisation of MAAs [52].

### 2.1. MAA UV-Absorbing Features 

In nature, various compounds are used for protection against the mutagenic effects of UV radiation. Dark pigment melanins found in humans and animals absorb the light in the UV and visible range. In humans, melanin protects skin by providing a physical barrier to UV and by absorbing 50%–75% of UVR [53]. Beyond UV absorbing properties, melanin pigments play a role as antioxidants [54]. In plants and many microorganisms, carotenoid pigments have a photoprotective role and are characterised by absorption in the range of 300–600 nm [13]. These pigments are important not only for the UV protection [4,17] but also in photosynthesis as an essential part of the photosynthetic apparatus [12,17].

Another well-known photoprotective compound is scytonemin, a small, yellow-brown, hydrophobic pigment that is found only in cyanobacteria [11]. Scytonemin absorbs in the UV-A wavelength region, with maximum absorption in purified form at 384 nm [55,56] and in vivo at 370 nm [20]. This pigment works not only as a potent UV protector but also as a powerful antioxidant [57] and has significant potential for sunscreen application in cosmetics [57,58]. Over a long period of history, cyanobacteria were exposed to extremely high UVR and consequently, these organisms developed additional protection of their DNA by applying two powerful UV-absorbing compounds, scytonemin and MAAs [10,59]. 

MAAs are the most common group of secondary metabolites found in aquatic species [17]. Marine and freshwater organisms utilising MAA compounds in UV protection include a variety of species from cyanobacteria, fungi, algae, to higher-order animals such as cnidaria, fishes, arthropods, mollusks, tunicates and echinoderms [7]. MAAs are involved in photoprotection due to their ability to absorb light in the range of UV-A (315–400 nm; making ~95 % of UV energy that penetrates the atmosphere) and UV-B (280–315 nm) without production of free radicals. The vast majority of different MAAs absorb within the UV-A range, like mycosporine-2-glycine, shinorine; porphyra-334 with ʎmax in the range of 332–334 nm (Figure 1B) [16]. Palythine, palythine-threonine, palythine-serine and palythinol have ʎmax at 320 nm, while palythine-serine sulfate and palythine-threonine sulfate have maximum absorbance at 321 nm. Usujirene, palythene; euhalothece-362 are characterised by the ability to absorb UV at higher UV-A wavelengths in the range of 357–362 nm [16].

MAAs containing cyclohexenone ring have maximum absorption within the UV-B range including mycosporine-glycine (ʎmax = 310 nm), mycosporine-taurine (ʎmax = 309 nm) and mycosporine-serine (ʎmax = 310 nm). On the other hand, MAA precursor 4-deoxygadusol is characterised by maximum absorption at 268 nm in acidic conditions and at 294 nm in basic environments [60]. Various amino acids in the MAA core result in different MAA profiles, which are characterised by different UV screening properties [60]. Therefore, only slight changes in the MAA structures will result in quick changes in the level of UV protection. This MAA property allows organisms to adjust to changeable UV conditions, which is important from the evolutionary perspective for survival under harsh and damaging UV radiation. Consequently, the accumulation of MAAs, like adaptable sunscreen, plays a critical role in the UV protection of marine organisms [8].

### 2.2. MAA Antioxidative Properties 

Oxidative stress happens due to the production of ROS, which includes, in general, the following products: hydrogen peroxide (H_2_O_2_), hydroxyl radical (OH^•^) and superoxide anion (O_2_^•−^). Prolonged exposure to sun radiation results in UV-induced oxidative stress. MAAs are ubiquitous metabolites that beyond their photoprotective role, also have a role as antioxidants [3,7,52]. MAAs are able to scavenge ROS and suppress singlet oxygen-induced damage [24,61,62,63]. An overview of the antioxidant properties of different MAAs has been provided in recent reviews [52,60]. In addition, a number of studies are summarised in Table 1 that assessed the antioxidative, anti-inflammatory and antiaging activities of individual MAAs in vitro.

Large differences in antioxidative capacities were reported for different MAAs. *In vitro* analyses of MAA antioxidative activities were done using different assays and it was apparent that modification in the external environment, such as the acidity or temperature, may increase their antioxidative properties [52]. Weak antioxidative activity was described for shinorine and porphyra-334 [24], although changes in the environment, like heat stress, resulted in an increase in the antioxidative properties for porphira-343 [67]. On the other hand, a highly abundant primary *MAA* mycosporine-glycine demonstrated effective antioxidant properties [24], as well as 4-deoxygadusol [16,72]. The antioxidant capacity of mycosporine-glycine was initially tested using peroxidation assay [24]. The concentration-dependent inhibition of lipid peroxidation was reported as a result of antioxidant action of this MAA [24]. The highest antioxidant activity was reported for mycosporine-glycine isolated from the marine lichen *Lichina pygmaea* at pH 8.5, which was eightfold higher than for ascorbic acid [61]. Furthermore, using the extracts of marine green alga, antioxidant properties were confirmed for mycosporine-glycine via the 2,2-diphenyl-1-picryhydrazyl (DPPH) assay, but not for porphyra-334 and shinorine [31]. On the contrary, Gacesa and colleagues [64] recently demonstrated antioxidant properties of porphyra-334 and shinorine using different free-radical quenching assays. The in vitro antioxidant activities were tested using the DPPH assay and were lower in two analysed MAAs compared to ascorbic acid [64]. The second assay, an oxygen radical absorption capacity (ORAC) assay, actually demonstrated substantial antioxidant activities for both MAAs [64]. Porphyra-334 and shinorine were also shown as the activator of the cytoprotective pathway demonstrating the potential for treating human degenerative diseases related to aging [64]. Specifically, the antioxidative activities of the two MAAs were related to the Keap1-Nrf2 pathway, which regulates cytoprotective cellular responses during oxidative stress. The Kelch-like ECH-associated protein 1 (Keap1) actin protein was found to detect changes in the redox status within the cell by controlling the activity of transcription nuclear factor erythroid 2-related factor 2 protein (Nrf2). Under oxidative stress, Nrf2 is activated due to detachment from Keap1 and Nrf2 was shown to regulate the expression of genes involved in the antioxidant response (antioxidant response element: ARE) and to play a role in oncogenesis [73]. In primary skin fibroblast cells, MAAs porphyra-334 and shinorine were able to provide protection from UVR-induced oxidative stress via the activation of the Keap1-Nrf2-ARE pathway and plus directly, by quenching free radicals [64]. Another MAA, mycosporine-2-glycine, in both in vivo and in vitro studies, demonstrated a high antioxidant activity that was equivalent to ascorbic acid [69]. To induce oxidative stress, the macrophage cells were exposed H_2_O_2_, while the presence of mycosporine-2-glycine resulted in downregulation of the expression of oxidative stress-induced genes such as Cu/Zn-*superoxide dismutase* 1 (*Sod1*) and *catalase* (*Cat*) [70].

The prevention of UV-induced damage and oxidative stress in epidermal skin cells involves the role of an endogenous defense system [74]. Antioxidant enzymes that play a critical role in this endogenous defense system include superoxide dismutase, catalase, glutathione peroxidase, glutathione reductase and thioredoxin oxidase enzymes. However, the cellular defense due to accumulated UV-absorbing compounds has been shown to be the first line of defense from oxidative stress [75]. In two scleractinian corals, the antioxidant capacities of MAAs were highly important as the protection coming from MAAs occurred before the action of antioxidative enzymes such as superoxide dismutase and catalase [75]. Thanks to the MAAs’ capacity to quench ROS and scavenge free radicals, UV induced damage is minimised in these vulnerable aquatic species. Consequently, the antioxidant defense mechanisms and ROS scavenging are critical for organisms’ survival [52].

### 2.3. MAA Anti-Inflammatory and Antiaging Properties 

The ROS production happening due to UV damage may lead to inflammation and immune stress responses [76]. Free radicals are able to work as signalling molecules changing gene expression, leading to oxidative stress, protein oxidation and resulting in the activation of inflammatory processes through the activation of different cellular pathways [77]. Inflammatory processes induced by UV exposure are mainly regulated by nuclear factor kappa b (NF-κB) and include a number of signalling mediators such as nitric oxide (NO), inducible NO synthase (iNOS) tumor necrosis factor α (TNF- α), cyclooxygenase (COX-2) and cytokines (i.e., interleukins) [52]. 

The anti-inflammatory activity in the microalga MAA extracts was tested in vitro by Suh et al. [31]. The HaCAT cells (immortal human keratinocytes) were exposed to UV radiation and supplemented with an increased concentration of MAAs (0.03, 0.15, or 0.3 mM). Real-time qPCR was used to evaluate the changes in gene expression of the *COX-2* gene that is found to be elevated in the case of tissue inflammation. MAAs have shown different anti-inflammatory properties, with porphyra-334 having no effect on *COX-2* gene expression, while mycosporine-glycine and shinorine had the inhibitory effect on the expression of inflammation-related gene (i.e., *COX-2* gene) [31]. Similarly, mycosporine-2-glycine reduced the transcription of genes critical for the inflammatory signalling processes, *COX-2* and *iNOS* [70]. Anti-inflammatory properties of MAAs shinorine and porphyra-334 were tested in human myelomonocytic cells under inflammatory stimulation by lipopolysaccharide (LPS) [65]. Both MAAs stimulated NF-κB activity prior to LPS induction, while under LPS-induced conditions, shinorine increased the activity of transcription factor NF-κB in a dose-dependent manner. On the other hand, porphyra-334 reduced the activity of NF-κB and demonstrated anti-inflammatory action. The aqueous extracts of red algae *Hydropuntia cornea* and *Gracilariopsis longissima* containing the mixture of MAAs (palythine, asterina-330, shinorine, porphyra-334; palythinol) and other compounds were reported to actively induce the production of TNF-α and anti-inflammatory/pro-inflammatory cytokine interleukin-6 [78].

The skin aging process is happening as a result of collagen destruction and elastin content reduction [79]. Exposure to UV radiation increases the rate of skin aging via oxidative stress and DNA mutations [80,81,82]. As UV exposure leads to photoaging, the evaluation of MAAs antiaging property is also critical when assessing MAAs’ potential for use in cosmetics. Suh et al. [31] tested the expression of genes related to the skin aging processes (i.e., genes for procollagen c-endopeptidase enhancer and elastin). The mRNA levels of these UV-suppressed genes were elevated in the presence of all analysed MAAs. The UV-induced downregulation of another gene related to skin aging (involucrin) was suppressed in the presence of mycosporine-glycine and shinorine, but not porphyra-334. Additional studies also confirmed the potential of MAAs as anti-photoaging molecules [66,68,82]. Consequently, more studies are needed to extend the promising potential of MAAs for their application in skin-care products.

## 3. Fluctuation of the MAA Content in Aquatic Species

MAAs are multifunctional secondary metabolites that beyond the role in photoprotection and as antioxidants, also play a part in osmotic regulation, control of reproduction, as well as nitrogen reservoirs; an accessory light-harvesting pigment in photosynthesis [83]. In the soft coral species *Lobophytum compactum*, MAA accumulation was three times higher in eggs than in maternal tissue indicating MAA importance for larval survival [84]. MAA levels and diversity are affected by seasonal fluctuations [85]. UV radiation is a major factor influencing MAA accumulation and resulting in changes in organisms’ MAA profile [38]. Spectral variability and intensity were found to affect the synthesis of MAAs [17]. Blue light within photosynthetically active radiation (PAR) and UV-A lights stimulated the production of MAAs in free-living algae and Antarctic diatoms [86,87,88,89,90]. The positive effect of blue light and UV-A exposure was also reported in the red macroalga *Chondrus crispus* [91], while UV-B had a positive effect on the MAA level in *Nodularia* cyanobacteria [92]. In some corals, both UV levels and PAR were required for successful MAA production [93,94]. Host–microbe interactions are another important factor influencing the diversity of MAAs. Lower MAA levels and diversity were reported for symbiotic dinoflagellates when in cultures than were in symbiosis within the coral host [32,95]. 

Together with UVR, there are other factors that stimulate the production of MAAs such as variation in salinity conditions and changes in nutrient availability [84,96,97,98,99]. In the halotolerant cyanobacterium *Aphanothece halophytica*, the production of mycosporine-2-glycine was stimulated more by the high salinity condition than by UV-B stress. In addition, the upregulation of MAA biosynthetic genes was reported in parallel to the accumulation of mycosporine-2-glycine in response to salt stress [100]. When hypersalinity is present in the surrounding environment, it can result in cell dehydration and the production of ROS, leading to oxidative stress. Via the synthesis of MAAs under the salt stress condition, MAAs are considered to be osmotic solutes that are helping in osmotic regulation and reestablishing osmotic balance [83]. Furthermore, when aquatic species are challenged by cold environmental conditions, MAAs are found to act as osmotic protector [83].

Beyond salinity stress and UV radiation [83,98,100] other environmental stress factors were reported to influence MAA levels such as desiccation [101] and heat stress [6,75]. Seasonal changes in UV levels and temperature had a positive effect on MAA accumulation in phytoplankton and the copepod *Cyclops abyssorum tatricus* obtained from an alpine lake [102]. These different environmental stress factors contribute to oxidative stress and MAA accumulation. Consequently, the importance of MAAs and their role as antioxidants is strongly stimulated by changes in external environmental conditions [60]. 

## 4. Genes from the MAA Biosynthetic Pathway and Their Regulation

The shikimate pathway was the first pathway proposed to be responsible for MAA biosynthesis [6,94]. In this pathway, 3-dehydroquinate (DHQ) was used as a basis for the production of 4-deoxygadusol (4-DG), which is recognised as a direct precursor of MAAs, leading to the production of primary and secondary MAAs [8,96,103]. The second pathway was proposed by Balskus and Walsh [104] and suggested that synthesis of MAAs occurs via the pentose phosphate pathway and from another intermediate sedoheptulose 7-phosphate (SH 7-P) through the four-enzyme shinorine pathway. The role of this pathway in the synthesis of primary MAA shinorine in the cyanobacterium *Anabaena variabilis* ATCC 29413 was confirmed when the entire gene cluster (containing all four genes) was cloned and expressed in vitro using heterologous expression system in *Escherichia coli.* The 6.5-kb shinorine biosynthetic gene cluster included the following genes: *dehydroquinate synthas*e (*DHQS*), *O-methyltransferase* (*O-MT*), *adenosine triphosphate (ATP) grasp* and a *nonribosomalpeptide synthetase (NRPS)* [104]. Via this pathway, 4-DG was initially produced thanks to the activity of DHQS and O-MT, followed by the second part of the pathway resulting in the generation of mycosporine-glycine and then shinorine. The same gene cluster has been also found in dinoflagellates [104], as well as in the sperm samples of the coral *Acropora digitifera* [48]. In coral dinoflagellates, three genes from the shinorine biosynthetic gene cluster were reported including two forms of *O-MT* gene, *NRPS* and ATP grasp homolog genes [9]. The lack of one gene from the shinorine biosynthetic pathway encoding DHQS reported for *Symbiodiniaceae* [9] may be due to limited sequence coverage, but could be potentially explained by the importance of host factors for the MAA biosynthesis [32,94]. Similar to coral dinoflagellate, the three-gene shinorine pathway was also reported in dinoflagellate *Heterocapsa triquetra*, in which *4-deoxygadusol (DDG) synthase*-encoding gene was fused to the *O-MT*-encoding gene (Figure 3A) [4,105]. Furthermore, not all genes from the shinorine pathway (called the *mys* cluster) were identified in the 363 cyanobacterial genomes (Figure 3B) [106]. Huge genetic variability was observed among different cyanobacteria species, indicating that the formation of shinorine may happen due to the activity of different enzymes. The major genetic variability reported was in terms of presence or absence of two distinct enzymes NRPS (encoded by gene *mysE*; from *Anabaena*-type *mys* cluster) and D-Ala-D-Ala ligase (encoded by gene *mysD* from *Nostoc*-type *mys* cluster) in different cyanobacterial species [106]. Further variability was detected in terms of the presence of dehydrogenases or reductases and duplication of *ATP-grasp* genes [106,107]. Consequently, there are still considerable gaps in understanding the genetic diversity and regulatory mechanisms of the MAA biosynthetic pathways.

## 5. Potential Use of MAAs in Human Skin Protection

UV radiation can lead to substantial skin damage, particularly within epidermal cells, resulting in an increased incidence of skin cancer [108]. In humans, excessive exposure to UVR is associated with the development of over 95% of skin cancers [109]. The effect of UV radiation depends on the UV range, where high energy UVB usually leads to direct DNA damage and has a highly mutagenic and carcinogenic effect compared to lower energy UVA radiation [110]. Both UVA and UVB damage DNA directly and indirectly through oxidative stress [110]. A major mechanism leading to skin cancer occurs due to the UV-induced formation of cyclobutane pyrimidine dimers (CPDs) leading to DNA base damage [111]. The mechanisms of skin protection include mitigation strategies in reducing UV-induced damage, such as DNA repair mechanisms, the accumulation of antioxidants and UV-absorbing compounds [4]. In the human body, the pigment melanin helps in the protection from UV-induced damage due to its broad spectrum and its antioxidant activity [54]. However, there are two forms of melanin, the brown/black pigment eumelanin that plays a photoprotective role and the orange/yellow pigment pheomelanin that is considered to be photosensitizing [112]. Pheomelanin produces superoxide and nitric oxide resulting in the generation of ROS that can damage DNA [112]; as recently revealed, this damage even occurs in darkness, with increased CPD formation during the first 2–3 hours after UV exposure [113]. Also, in human skin, only 50%–75% of UVR is absorbed by melanin [53]. Subsequently, human skin needs additional protection via externally applied sunscreens. However, current chemical UVR protection is not adequate because these commercially available sunscreen products contain active ingredients that lack photostability, can produce free radicals leading to skin damage, irritation, skin aging and also can cause allergic reactions. Furthermore, the UV-filter compounds included in cosmetic products and used in the packing industry are disposed of, resulting in environmental pollution [114]. These organic compounds coming from wastewater reach groundwater and lead to the destruction of ecosystems such as coral reefs. Toxic effects of the sunscreens and specifically the UV-filter compound (*oxybenzone, benzophenone-3)*, have been reported on marine life, particularly on tested coral planulae and cultured cells [115]. Consequently, biodegradable solutions, such as nature-based UV-filters like MAAs, should be used instead of the current chemical UV-filters and may help in the prevention of further environmental damage. 

Interest in MAAs has been growing during the past decades. The use of MAAs for health products and cosmetics was recently reviewed by Chrapusta and colleagues [116]. MAAs are photostable, transparent compounds found in a wide range of aquatic species. As an old-new ecologically friendly option, MAAs show a stable photoprotective and antioxidative capacity against a range of UVR [117,118]. A number of research–based partnerships led to the generation of over 60 patents around the world [116]. The first application of MAAs, specifically shinorine, has been utilised in the product Helioguard^®^365 [119]. *In vitro* studies on the human keratinocyte cells confirmed the antiaging activity of the product when exposed to UV-A and stable activity of MAAs for a three-month period [119]. There are a few other products on the market that also utilise MAAs such as Helionori^®^ and an increasing number of industry patents [116]. Still, more can be done in the market still dominated by chemical UV sunscreen products by enhancing industry–research collaborations to speed up the technological advancement of the MAA application as organic sunscreens. 

## 6. Conclusions

Environmental pollution coming from the use of chemical UV filters is becoming an increasing problem that destabilizes the natural environment and contaminates groundwater. MAAs are excellent candidates for use in the cosmetic industry and for improved skin protection in an ecologically friendly way as water-soluble, colourless; highly diverse compounds. The MAAs’ ability to easily change form, as well as their UV properties and antioxidant capacities, present enormous biotechnological potential. The fact that MAA synthesis is adjustable in relation to changes in the environment, such as an increase in UV radiation, could be utilised further. Improving our understanding of the regulatory mechanisms and genetic diversity of MAA biosynthetic pathways will allow us to use MAAs in various biotechnological applications. Future research is needed to allow for better control of MAA biosynthesis and the production of adjustable, personalised natural sunscreens.

## Figures and Tables

**Figure 1 marinedrugs-17-00638-f001:**
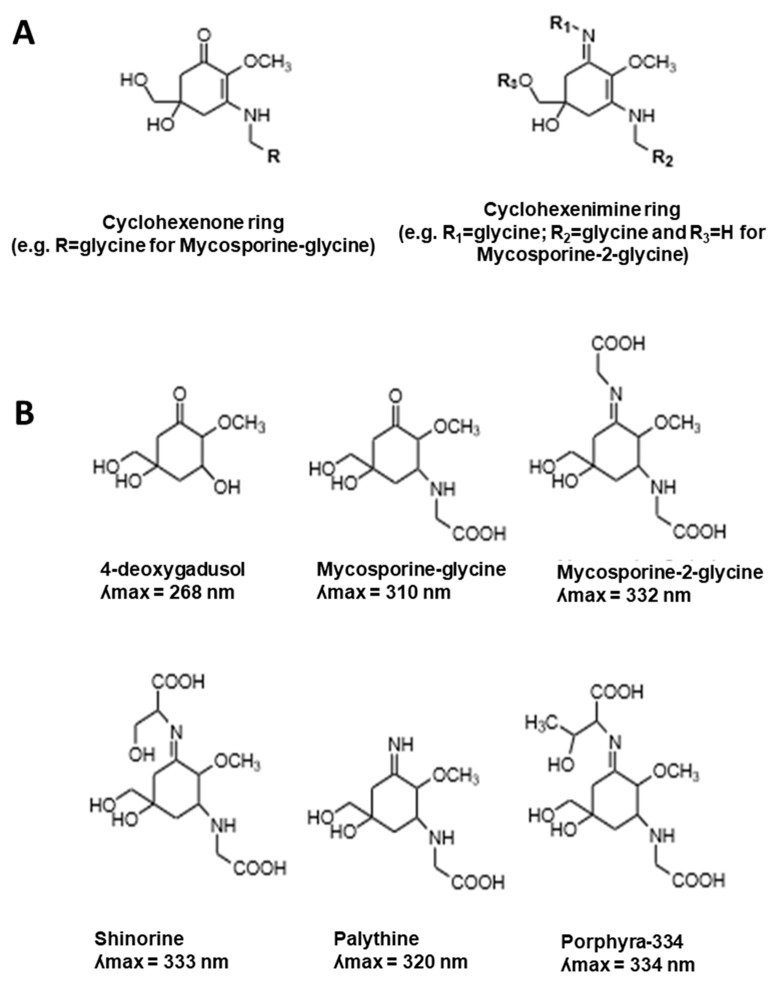
Chemical structures of mycosporine-like amino acids (MAAs): (**A**) MAA core composed of a cyclohexenone, a cyclohexenone, or cyclohexenimine ring conjugated to an amino acid residue or its imino alcohol; (**B**) MAA precursor 4-deoxygadusol, plus the primary MAAs found including mycosporine-glycine, mycosporine-2-glycine, shinorine, palythine and porphyra-334, including the maximum absorbance values.

**Figure 2 marinedrugs-17-00638-f002:**
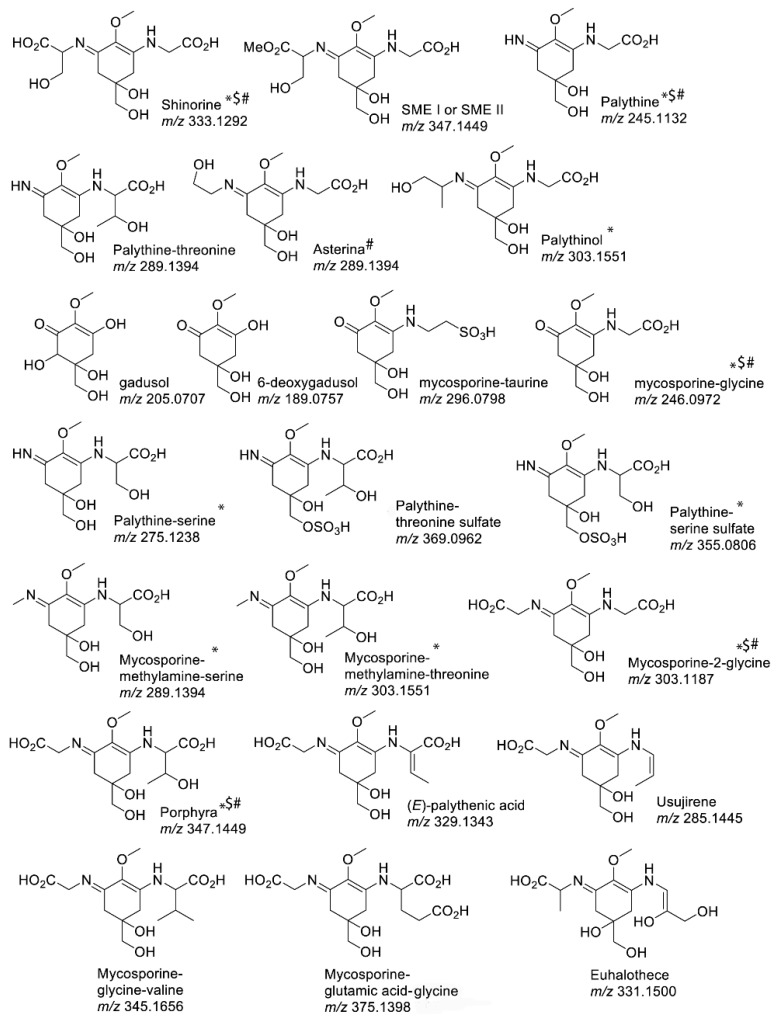
Chemical structures and masses (m/z) of commonly found MAAs in the red alga (indicated by #), *Symbiodiniaceae* (indicated by $) and the hermatypic coral *Stylophora pistillata* (indicated by *) as adapted from Rosic et al. [45].

**Figure 3 marinedrugs-17-00638-f003:**
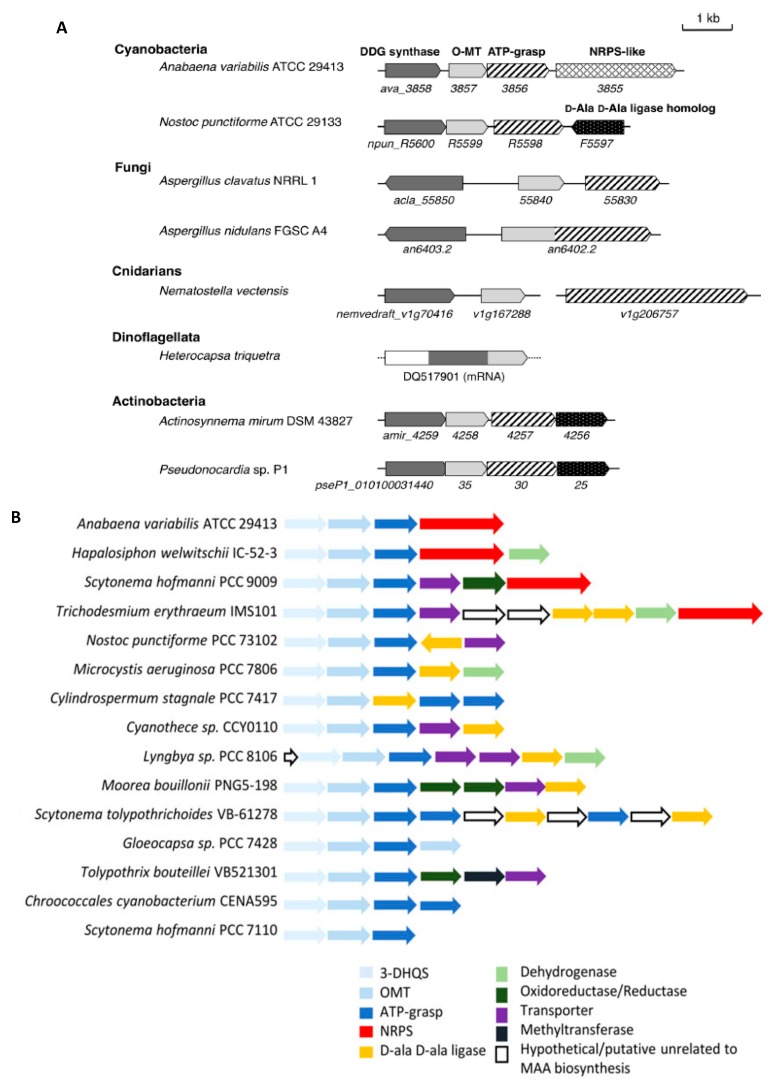
Gene clusters for mycosporines and MAAs biosynthesis. (**A**) The MAA gene clusters from eight different species including representatives from bacteria, algae, fungi; cnidarians (adapted from Miyamoto et al. [105]). (**B**) Gene clusters for MAA biosynthesis identified in different cyanobacterial species (adapted from reference [106]).

**Table 1 marinedrugs-17-00638-t001:** Properties of main MAAs based on in vitro studies using cell cultures exposed to individual MAAs isolated from various species.

MAAs	Activity: UV-Absorbing	Activity: Antioxidative	Activity: Anti- Inflammatory	Activity: Antiaging	Sources of MAAs and References
**Mycosporine-glycine**	Yes ʎmax = 310 nm	Yes [24,31,51,61,63]	Yes [31]	Yes [31]	Red alga *Porphyra tenera* [24] Green alga *Chlamydomonas hedleyi* [31] Ascidian *Lissoclinum patella* [63] Marine lichen *Lichina pygmaea* [61]
**Shinorine**	Yes ʎmax = 333 nm	Yes [24,51,64] No [31]	Yes [31] No [65]	Yes [31,66]	Red alga *Chondrus yendoi* [64] Red alga *Porphyra* sp. [65] Green alga *Chlamydomonas hedleyi* [31] Red algae *Porphyra* sp. & *Palmaria palmate* [66]
**Porphyra-334**	Yes ʎmax = 334 nm	Yes [24,51,64,67,68] No [31]	Yes [65,68] No [31]	Yes [31,66] No [31]	Green alga *Chlamydomonas hedleyi* [31] Red algae *Porphyra* sp. & *Palmaria palmate* [66] Red alga *Porphyra yezoensis* [68]
**Mycosporine-2-glycine**	Yes ʎmax = 332 nm	Yes [51,69,70]	Yes [70]	Yes [70]	Cyanobacterium *Aphanothece halophytica* [51,69,70]
**Palythine**	Yes ʎmax = 320 nm	Yes [71] No [69]	-	Yes [66]	Red alga *Chrondus yendoi* [71] Cyanobacterium *Aphanothece halophytica* [69] Red algae *Porphyra* sp. & *Palmaria palmate* [66] Cyanobacterium *Aphanothece halophytica* [69]

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
