# Peer review of "Mycosporine-Like Amino Acids: Making the Foundation for Organic Personalised Sunscreens"

_marinedrugs, 2019, doi:10.3390/md17110638_

Round 1

Reviewer 1 Report

The manuscript entitled "Mycosporine-like Amino Acids: making the foundation for organic personalized sunscreens" is well constructed and deals with a topic of current interest and of considerable importance in the field of research.

The introduction is sufficiently consistent with the objectives of the review, although it may be extended by taking into account certain aspects that are adequately structured in the paragraphs in which the manuscript is divided.

Since this is a review, however, it would be appropriate to supplement the manuscript with a synoptic table that gives a general picture of all the properties of the different MAA (or at least of the main ones also mentioned in the body of the text), with an appropriate bibliography to support it. Recent literature also reports several MAAs in elasmobranches that should be considered by the Author. These and those of several other marine organisms should not only be discussed, but of course also included in the synoptic table mentioned above.

Overall, however, the work is absolutely interesting and, once the missing parts have been integrated and enriched with the recommended in-depth studies, but above all with the insertion of the table, which could also be attached as additional material, the manuscript will acquire considerable dignity and importance.

In view of everything I have said, I believe that additions and changes are necessary, but they can be considered partial revisions, not particularly demanding, but which cannot be ignored in order to publish the manuscript.

Author Response

Dear Reviewer,

Thanks a lot for the positive feedback on the manuscript. The response to your comments is provided below.

Kind regards,

Nedeljka Rosic

Reviewer 1

Comments and Suggestions for Authors

The manuscript entitled "Mycosporine-like Amino Acids: making the foundation for organic personalized sunscreens" is well constructed and deals with a topic of current interest and of considerable importance in the field of research.

The introduction is sufficiently consistent with the objectives of the review, although it may be extended by taking into account certain aspects that are adequately structured in the paragraphs in which the manuscript is divided.

Since this is a review, however, it would be appropriate to supplement the manuscript with a synoptic table that gives a general picture of all the properties of the different MAA (or at least of the main ones also mentioned in the body of the text), with an appropriate bibliography to support it.

Recent literature also reports several MAAs in elasmobranches that should be considered by the Author. These and those of several other marine organisms should not only be discussed, but of course also included in the synoptic table mentioned above.

Respond: This part has been extended. In addition, a  number of studies, which assessed in vitro the anti-oxidative, anti-inflammatory and anti-aging activities of individual MAAs are summarised within Table 1 including MAA sources and references. Furthermore, additional MAAs discovered recently were included in the text of the manuscript.

Lines 62-64: A huge variety of MAAs have been reported in cyanobacteria [20-22], red algae [23-28], fungi [29], green algae [30, 31], dinoflagellates [32-34], invertebrates (e.g. sponges [15, 35, 36], corals [21, 37, 38], sea urchins [39, 40]) and vertebrates such as fish [41-44].

Lines 120-122: In addition, two new MAAs (LC-343 and mycosporine-ethanolamine) along with well-known asterina-330 and shinorine were recently isolated from the marine sponge Lendenfeldia chondrode [36].

Lines 125-126: However, beyond the diet, de novo synthesis of gadusol (the MAA precursor) was reported in some fish [43, 44] and corals [48].

Reviewer 2 Report

The manuscript „Mycosporine-like Amino Acids: making the foundation for organic personalized sunscreens” by Nedeljka N. Rosic is a very thoroughly prepared review based on extensive literature data. However, in my opinion it should contain also some additional information.

Lines 161-171. in the context of MAA antioxidant activity, author should also describe Nrf2 activity (especially if studies on Nrf2 activity are cited: e.g. point 50). Lines 177-183. No information about NFkB and TNFa interactions with MAA. Are there any information on MAA effects on proliferation, differentiation on cancer transformation? Mainly in the context of skin cancer. Minor corrections: Line 137. “hydrogen peroxide (H2O2), hydroxyl radicals (OH) and superoxide anion.” OH is not a hydroxyl radical, lacking the chemical formula of superoxide anion.

Author Response

Dear Reviewer,

Thanks a lot for the positive feedback on the manuscript. The response to your comments is provided below.

Kind regards,

Nedeljka Rosic

Reviewer 2

The manuscript „Mycosporine-like Amino Acids: making the foundation for organic personalized sunscreens” by Nedeljka N. Rosic is a very thoroughly prepared review based on extensive literature data. However, in my opinion, it should contain also some additional information.

Lines 161-171. in the context of MAA antioxidant activity, author should also describe Nrf2 activity (especially if studies on Nrf2 activity are cited: e.g. point 50).

Lines 177-183. No information about NFkB and TNFa interactions with MAA. Are there any information on MAA effects on proliferation, differentiation on cancer transformation? Mainly in the context of skin cancer.

Respond: This part has been extended and the mechanisms have been explained.

Lines 163-165: The highest antioxidant activity was reported for mycosporine-glycine isolated from the marine lichen Lichina pygmaea at pH 8.5, which was 8-fold higher than for ascorbic acid [61].

Lines 174-187: Specifically, the antioxidative activities of the two MAAs were related to the Keap1-Nrf2 pathway, which regulates cytoprotective cellular responses during oxidative stress. The Kelch-like ECH-associated protein 1 (Keap1) actin protein was found to detect changes in the redox status within the cell by and control the activity of transcription nuclear factor erythroid 2-related factor 2 protein (Nrf2). Under oxidative stress, Nrf2 is activated due to detachment from Keap1 and Nrf2 was shown to regulate the expression of genes involved in the antioxidant response (antioxidant response element: ARE) and to play a role in oncogenesis [67]. In primary skin fibroblast cells, MAAs porphyra-334 and shinorine were able to provide protection from UVR-induced oxidative stress via the activation of the Keap1-Nrf2-ARE pathway and plus directly, by quenching free radicals [66]. Another MAA, mycosporine-2-glycine, in studies done in vivo and in vitro demonstrated a high antioxidant activity that was equivalent to ascorbic acid [68]. In macrophage cell line exposed H2O2, to induce oxidative stress, the presence of mycosporine-2-glycine resulted in down-regulation of the expression of oxidative stress-induced genes such as superoxide dismutase 1 and catalase [69].

Lines 203-206: Inflammatory processes induced by UV exposure are mainly regulated by nuclear factor kappa b (NF-κB) and include a number of signaling mediators such as nitric oxide (NO), inducible NO synthase (iNOS) tumor necrosis factor α (TNF- α), cyclooxygenase (COX-2) and cytokines (i.e. interleukins) [52].

Lines 214-223: Similarly, mycosporine-2-glycine reduced the transcription of genes critical for the inflammatory signaling processes, COX2 and iNOS [69]. Anti-inflammatory properties of MAAs shinorine and porphyra-334 were tested in human myelomonocytic cells under inflammatory stimulation by lipopolysaccharide (LPS) [74]. Both MAAs stimulated NF-κB activity prior to LPS induction, while under LPS-induced conditions, shinorine increased the activity of transcription factor NF-κB in a dose-dependent manner. On the other hand, porphyra-334 reduced the activity of NF-κB and demonstrated anti-inflammatory action. The aqueous extracts of red algae Hydropuntia cornea and Gracilariopsis longissima containing the mixture of MAAs (palythine, asterina-330, shinorine, porphyra-334, and palythinol) and other compounds was reported to actively induce the production of TNF-α and anti-inflammatory/pro-inflammatory cytokine interleukin-6 [75].

Minor corrections: Line 137. “hydrogen peroxide (H2O2), hydroxyl radicals (OH) and superoxide anion.” OH is not a hydroxyl radical, lacking the chemical formula of superoxide anion.

Respond: Thanks to the reviewer’s feedback. This has been fixed and appropriate formulas have been provided.

Lines 146-147: Oxidative stress happens due to the production of ROS, which includes, in general, the following products: hydrogen peroxide (H2O2), hydroxyl radical (OH•) and superoxide anion (O2•−).

Round 2

Reviewer 1 Report

I believe that the author has made all the corrections and additions suggested enriching the manuscript in a remarkable way. The author has also added an exhaustive table of the properties of the compounds as suggested in the review, carefully accompanied by an adequate bibliography. For these reasons I believe that the manuscript in this form can be considered definitive and ready for publication.